Citation: *Molecular Systems Biology* 9:662
www.molecularsystemsbiology.com

# Characterization of drug-induced transcriptional modules: towards drug repositioning and functional understanding

Murat Iskar[1], Georg Zeller[1], Peter Blattmann[2,3], Monica Campillos[4,5], Michael Kuhn[6], Katarzyna H Kaminska[1,9], Heiko Runz[3,7], Anne-Claude Gavin[1], Rainer Pepperkok[2,3], Vera van Noort[1] and Peer Bork[1,8,*]

[1] Structural and Computational Biology Unit, European Molecular Biology Laboratory (EMBL), Heidelberg, Germany, [2] Cell Biology/Biophysics Unit, EMBL, Heidelberg, Germany, [3] Molecular Medicine Partnership Unit (MMPU), EMBL, University of Heidelberg, Heidelberg, Germany, [4] Institute for Bioinformatics and Systems Biology, Helmholtz Center Munich–German Research Center for Environmental Health (GmbH), Neuherberg, Germany, [5] German Center for Diabetes Research (DZD), Neuherberg, Germany, [6] Biotechnology Center, TU Dresden, Dresden, Germany, [7] Institute of Human Genetics, University of Heidelberg, Heidelberg, Germany and [8] Max-Delbrück-Centre for Molecular Medicine, Berlin, Germany
[9]Present address: International Institute of Molecular and Cell Biology in Warsaw, ul. Ks. Trojdena 4, 02-109 Warsaw, Poland
* Corresponding author. Structural and Computational Biology Unit, European Molecular Biology Laboratory (EMBL), Meyerhofstrasse 1, Heidelberg, Germany.
Tel.: +49 6221 387 8526; Fax: +49 6221 387 8517; E-mail: bork@embl.de

In pharmacology, it is crucial to understand the complex biological responses that drugs elicit in the human organism and how well they can be inferred from model organisms. We therefore identified a large set of drug-induced transcriptional modules from genome-wide microarray data of drug-treated human cell lines and rat liver, and first characterized their conservation. Over 70% of these modules were common for multiple cell lines and 15% were conserved between the human *in vitro* and the rat *in vivo* system. We then illustrate the utility of conserved and cell-type-specific drug-induced modules by predicting and experimentally validating (i) gene functions, e.g., 10 novel regulators of cellular cholesterol homeostasis and (ii) new mechanisms of action for existing drugs, thereby providing a starting point for drug repositioning, e.g., novel cell cycle inhibitors and new modulators of α-adrenergic receptor, peroxisome proliferator-activated receptor and estrogen receptor. Taken together, the identified modules reveal the conservation of transcriptional responses towards drugs across cell types and organisms, and improve our understanding of both the molecular basis of drug action and human biology.
*Molecular Systems Biology* **9**: 662; published online 30 April 2013; doi:10.1038/msb.2013.20
*Subject Categories:* computational methods; molecular biology of disease
*Keywords:* cell line models in drug discovery; drug-induced transcriptional modules; drug repositioning; gene function prediction; transcriptome conservation across cell types and organisms

## Introduction

Understanding the complex responses of the human body to drug treatments is vitally important to address the efficacy and safety-related issues of compounds in later stages of drug development and, thus, to reduce high attrition rates in clinical trials (Kola and Landis, 2004). The fundamental challenge towards this goal lies in the selection and thorough characterization of model systems that can accurately recapitulate the drug response of human physiology for diverse drug-screening projects (Jones and Diamond, 2007; Sharma *et al*, 2010; Dow and Lowe, 2012).

One way of obtaining unbiased, large-scale readouts from model systems is genome-wide expression profiling of the transcriptional response to various drug treatments (Feng *et al*, 2009; Iskar *et al*, 2011). This has first been systematically explored in model organisms, such as *Saccharomyces cerevisiae*, with the aim to elucidate drug mechanism of action

(MOA) based on their transcriptional effects (Hughes *et al*, 2000; Ihmels *et al*, 2002; di Bernardo *et al*, 2005). Simultaneously, coexpression analysis and transcriptional modules of the yeast data allowed the inference of functional roles for genes that respond coherently to these perturbations (Hughes *et al*, 2000; Ihmels *et al*, 2002; Wu *et al*, 2002; Segal *et al*, 2003; Tanay *et al*, 2004).

Recently, the Connectivity Map (CMap) successfully extended the concept of large-scale gene expression profiling of drug response to human cell lines (Lamb *et al*, 2006; Lamb, 2007). In parallel, drug-induced expression changes have been profiled at a large scale in animal models, such as rat liver (Ganter *et al*, 2005; Natsoulis *et al*, 2008). Computational advances in mining these data have improved signature comparison methods leading to novel drug–drug (Subramanian *et al*, 2005; Lamb *et al*, 2006; Iorio *et al*, 2009, 2010) and drug–disease (Hu and Agarwal, 2009; Sirota *et al*,

2011; Dudley *et al*, 2011; Pacini *et al*, 2012) connections based on their (anti)correlated transcriptional effects (Qu and Rajpal, 2012; Iorio *et al*, 2012). However, these mammalian transcriptional readouts still need to be utilized for uncovering the underlying gene regulatory networks and for predicting gene function and delineating pathway membership. Along those lines, we used a biclustering approach that is well-suited for revealing the modular organization of transcriptional responses to drug perturbation (Ihmels *et al*, 2002; Prelić *et al*, 2006), as it can group coregulated genes with the drugs they respond to (technically, each bicluster consists of both a gene and a drug subset). We applied it to large-scale transcriptome resources for three human cell lines and rat liver to generate, for the first time, a large compendium of mammalian drug-induced transcriptional modules. We extensively characterized these modules in terms of functional roles of the genes and the bioactivities of the drugs they contain, in order to gain insights into both, drug MOA and the perturbed cellular systems (Figure 1).

Comparing drug-induced transcriptional modules generated independently for each of the three human cell types and rat liver allowed us to assess their conservation across tissue types and organisms. Although it has been noted earlier that particular transcriptional changes along developmental trajectories, in different growth conditions, stress or disease can be conserved between species (Stuart *et al*, 2003; van Noort *et al*, 2003), it remains an open question, how well the modular organization of the transcriptome is conserved across tissues and organisms (Miller *et al*, 2010; Zheng-Bradley *et al*, 2010; Dowell, 2011). In this context, our results on the conservation of drug-induced transcriptional modules contribute to a better understanding of the degree to which cell line models recapitulate the biological processes and signaling pathways taking place in the human physiological context (Jones and Diamond, 2007; Sharma *et al*, 2010).

## Results and discussion

### Identification of drug-induced modules in human cell lines and rat liver

To identify and compare drug-induced transcriptional modules from human cell lines as well as rat liver tissue, we exploited data from the following two public resources: (i) the CMap (Lamb *et al*, 2006), which contains 6100 expression profiles of several human cancer cell lines treated with 1309 drug-like small molecules (hereafter simply referred to as 'drugs') (Supplementary Figure 1) and (ii) the DrugMatrix resource, a large data set of 1743 expression profiles from liver tissue of drug-treated rats (Natsoulis *et al*, 2008). Raw microarray data were subjected to quality control and preprocessing procedures to improve data consistency and reduce batch effects (Iskar *et al*, 2010). For CMap, this resulted in a usable set of expression measurements of 8964 genes in three human cell lines (HL60, MCF7 and PC3, see Materials and Methods), each treated with the same set of 990 drugs. From the rat data set, only genes with orthologous human genes present in CMap were considered. This yielded expression profiles for 3618 genes in response to treatments with 344 distinct drugs (Supplementary Figure 1).

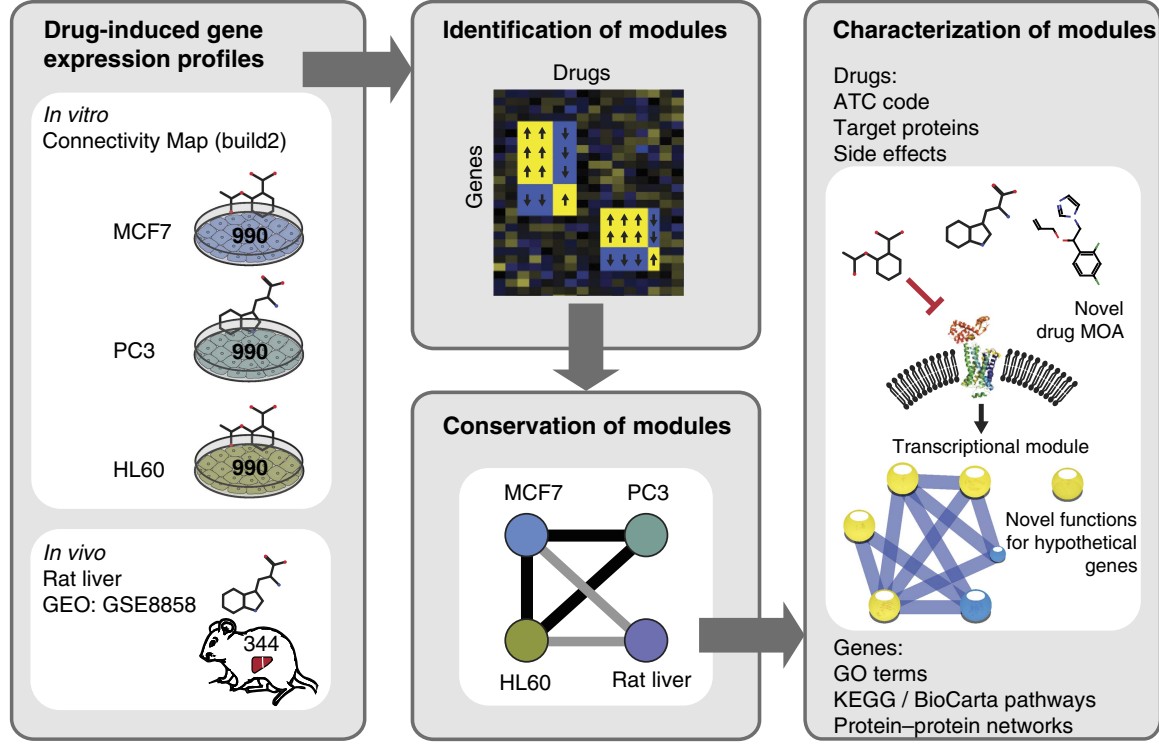

**Figure 1** Overview of the method. Workflow to identify and characterize drug-induced transcriptional modules across four microarray data sets from three human cancer cell lines and rat liver. Drug and gene sets of conserved drug-induced transcriptional modules (CODIMs) were characterized in detail using several annotation resources. These reliable CODIMs allow us to propose new MOA for marketed drugs and novel biological roles for poorly characterized genes which were validated experimentally.

  

Drug-induced transcriptional modules were detected and tested for statistical significance separately in each of the four matrices of expression data using an unsupervised biclustering approach that has previously been shown to maintain high accuracy even with noisy input data (Iterative Signature Algorithm (ISA); Bergmann *et al*, 2003; Ihmels *et al*, 2004; Prelić *et al*, 2006). Each resulting bicluster (hereafter referred to as 'module') consists of a subset of genes and a subset of drugs that coherently regulate these genes. To ensure comprehensiveness and robustness of this analysis, we explored a wide range of parameter settings for the ISA workflow (Supplementary Figure 2 and Supplementary Tables 1 and 2), and in addition, applied a size threshold to exclude small, likely spurious modules as motivated by previous studies (Langfelder and Horvath, 2008, and see Materials and Methods).

As a result, we identified robust, drug-induced transcriptional modules in each system individually: 25 in MCF7 cells, 28 in PC3 cells, 29 in HL60 cells and 43 in rat liver (Figure 2A and Supplementary Data set 1). On average, these modules contain 70 genes in the human cell lines and 50 genes in the rat liver, induced by an average of 29 drug treatments in both data sets. Within each data set, modules can overlap (on average, 7% of the genes and 34% of the drugs were contained in

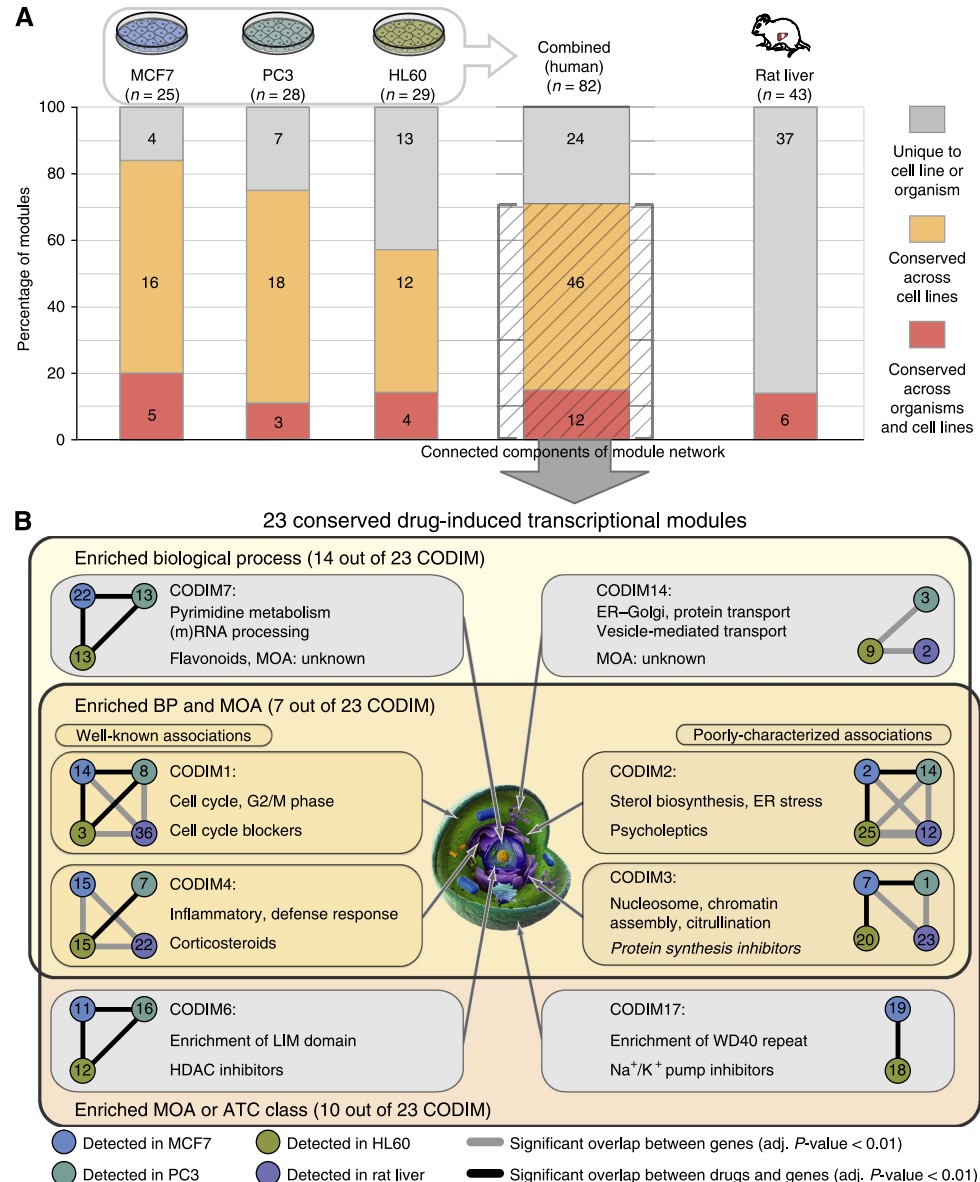

**Figure 2** Conservation of drug-induced transcriptional modules. (**A**) The number and proportion of transcriptional modules identified separately in each human cell line and rat liver that are codetected in multiple cell lines and/or organisms (gene overlap of modules were deemed significant with Fisher's exact test, FDR-corrected *P*-value < 0.01). Twenty-three CODIMs were defined from the connected components of the module network (using reciprocal best-hits only). (**B**) Functional characterization of conserved drug-induced modules. Connected modules (with data set-specific labels), enriched biological process (in yellow set, 61% of CODIM) and characteristic compound MOA (in red set, 44% of CODIM, in italics) of selected CODIMs are shown (BP, biological process; HDAC, histone deacetylase; WD40 repeat, β-transducin repeat). Graph inlets: conservation of modules across cell lines and species as measured by overlapping gene and drugs (Supplementary Figure 3).

multiple modules), reflecting the effects of polypharmacological drugs that modulate multiple targets (Hopkins *et al*, 2006; Keiser *et al*, 2009), which in turn can lead to perturbation of multiple pathways by the same drug.

## Conservation of drug-induced modules across cell types and organisms

As appropriate animal models that accurately recapitulate the human drug response are crucial in drug discovery and development, we assessed the conservation of drug responses at the transcriptome level by comparing drug-induced transcriptional modules across cell lines and organisms. This resulted in a network of module similarity calculated using a reciprocal best-hit approach, which linked modules from different cell lines and rat liver to each other if their gene members significantly overlapped between data sets (Supplementary Table 3). In addition, we assessed the drug overlap of modules linked across cell lines (see Materials and Methods). For 58 out of 82 modules (71%) from human cell lines, we identified a corresponding module in at least 1 other cell line, which yielded a total of 23 nonredundant, conserved drug-induced transcriptional modules (CODIMs; Figure 2A and Supplementary Figure 3). A conservation level of 71% between cell lines is in line with a previous study that assessed the conservation of constitutive coexpression networks across human brain regions (Oldham *et al*, 2008). In addition to similarity across cell lines, we found considerable (statistically significant) conservation of human modules in rat liver ranging from 3 to 5 out of 25 to 29 individual modules per cell line (15% average ratio, permutation-based *P*-value <0.001; Figure 2A and Supplementary Figures 3 and 4). Conservation of drug-induced modules across cell lines and rat liver was observed to be robust to the number of drug experiments or intrinsic parameters of the ISA procedure as long as drug perturbations were sufficiently diverse (Supplementary Figure 5), even though the rat liver modules are based on much fewer genes (3618 orthologs) and drugs (141 in common) than the cell line data (Supplementary Figure 1). In the context of the debate about the conservation of transcriptional regulatory networks between organisms (Liao and Zhang, 2006; Miller *et al*, 2010; Zheng-Bradley *et al*, 2010; Brawand *et al*, 2011), our assessment of 15% conservation of drug-induced modules across species should be considered as a lower limit; the divergence of modules can not only be attributed to completely different pharmacokinetics between the human *in vitro* model and the rat *in vivo* system, but are likely also the result of different tissue origins (Liao and Zhang, 2006; Zheng-Bradley *et al*, 2010) and heterogeneity of compared data sets, e.g., distinct drug perturbations profiled and technical differences between microarray protocols and platforms, with a limited set of orthologous probes/genes.

## Characterization of gene and drug members of drug-induced modules

To verify the functional coherence of drug-induced transcriptional modules, we first compared their gene members with functional association networks of human and rat provided by STRING (Szklarczyk *et al*, 2011). We observed that 12 out of 23 CODIMs were significantly enriched in functionally related genes (permutation-based *P*-value <0.05; 30 out of 82 individual modules from human cell lines and 13 out of 43 from rat liver; Supplementary Figure 6). In agreement with the earlier functional analyses of coexpression modules (Oldham *et al*, 2008), this indicates that drug-induced modules are useful to infer gene function.

We further characterized CODIMs with respect to both the functional annotations of gene members and the biological effects of each of the associated drugs (see Materials and Methods for details). This analysis revealed that CODIMs cover a broad diversity of cellular processes in response to drug treatments (Figure 2B and Supplementary Tables 4 and 5). On the basis of these module annotations, we next assessed whether gene annotations and drug indications were consistent in the light of the current biological and pharmacological understanding of these processes (Figure 2B and Supplementary Table 4). In a few cases, the enrichment of MOAs and biological processes in CODIMs agreed well, implying a known mechanism of the transcriptional response. For example, in CODIM1 the majority of drugs are known as cell cycle blockers and the corresponding gene annotations of this module are enriched in 'cell cycle and the transition to M phase' (Crawford and Piwnica-Worms, 2001; Whitfield *et al*, 2002, and see Figure 2B and Supplementary Tables 4 and 5 for additional examples). However, for the majority of modules the connection between enriched terms of genes and drugs was less obvious, suggesting novel aspects of drug MOAs; CODIM7 is enriched for genes functioning in RNA processing and pyrimidine metabolism, while it is associated with a heterogeneous set of chemicals containing several flavonoids and alkaloids. The impact of these natural products on the human organism is not fully understood, but there is evidence that some of these chemicals, including quercetin and kampferol, inhibit RNA synthesis (Nose, 1984; Kanakis *et al*, 2006, 2007; Nafisi *et al*, 2010; Supplementary Tables 4 and 5).

Another unexpected connection was observed for the highly conserved CODIM2; functional analysis of its gene members indicates a role in (chole)sterol biosynthesis and endoplasmic reticulum (ER) related processes, while the associated drugs were mostly psychiatric medications in CMap (Anatomical Therapeutic Chemical (ATC) code N05—psycholeptics), including many antipsychotics (e.g., chlorpromazine) and antidepressants (e.g., fluoxetine); whereas in the rat liver, all associated drugs were statins (cholesterol-lowering drugs; Supplementary Table 5). A broad regulatory effect of diverse psycholeptic drugs on cholesterol biosynthesis was unexpected, although it has been reported for a few cases, and inhibition of low-density lipoprotein (LDL)-derived cholesterol transport to the ER was proposed as an underlying mechanism (Fernø *et al*, 2005; Kristiana *et al*, 2010; Canfran-Duque *et al*, 2012). This association also offers a potential explanation for some of the observed side effects of drugs enriched in CODIM2 (e.g., galactorrhea or breast enlargement, see Supplementary Table 5), reflecting the phenotypic consequences of dysregulation of steroid hormone biosynthesis. Taken together, these findings highlight that drug-induced modules and their annotations provide hints about complex,

potentially novel biological phenomena of drug responses and generate hypotheses for further experimental investigations.

## Functional discovery within drug-induced modules

CODIMs can be utilized to functionally characterize the genes therein, in particular poorly annotated ones. For example, CODIM2, which is conserved across all cell lines and rat liver

shows a clear enrichment for (chole)sterol biosynthesis and regulation (see above, Supplementary Table 5). The biological processes underlying cholesterol homeostasis are fundamental to cellular metabolism and its regulation at the transcriptional level has been studied widely (Brown and Goldstein, 2009); thus, we systematically studied the genes of this module. Grouping the expression patterns in CODIM2 with hierarchical clustering revealed two major submodules (Figure 3A). Analysis of the respective genes in the context of a protein-interaction network (obtained from STRING

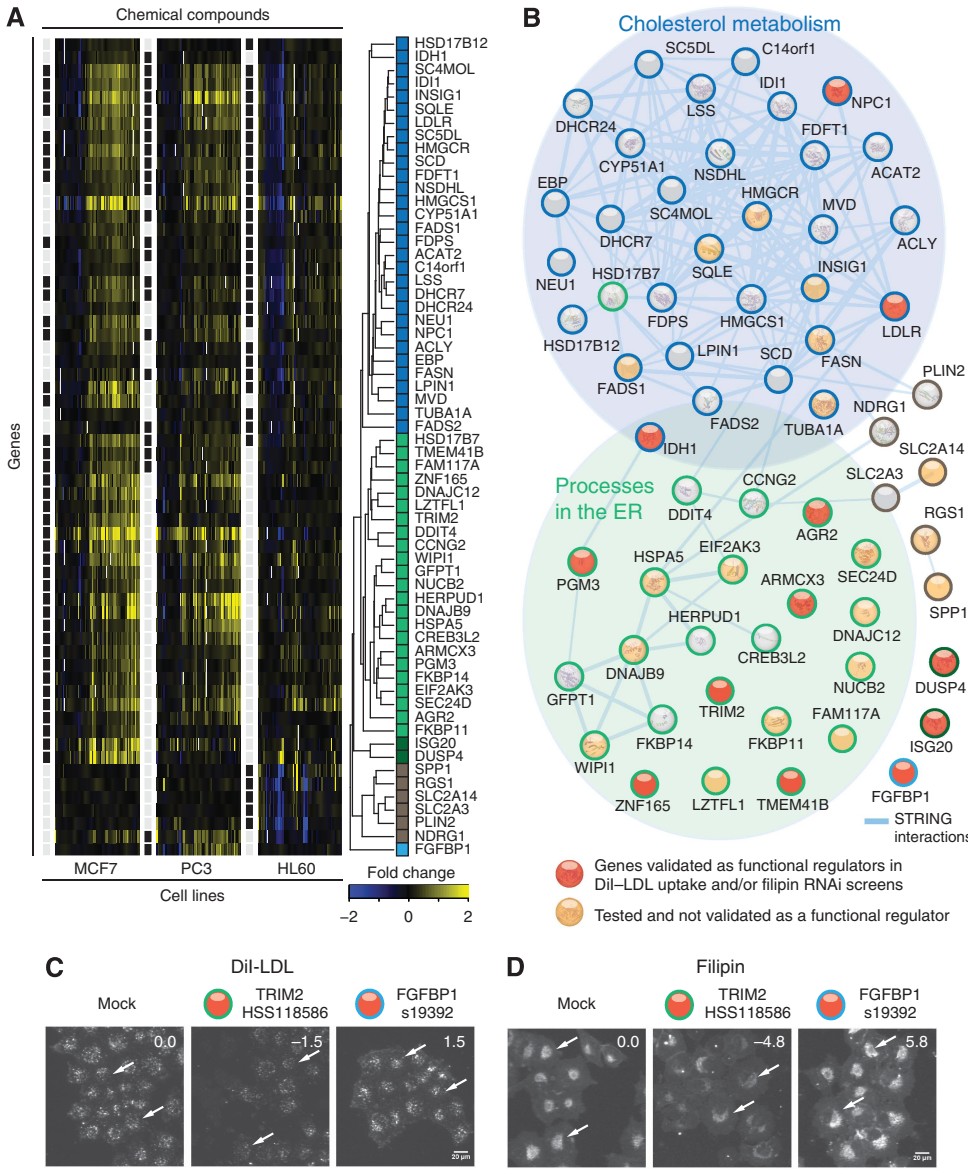

**Figure 3** Novel functional roles of uncharacterized genes as functional regulators of cellular cholesterol levels. (**A**) Heatmap of drug-induced gene expression changes (fold change) of CODIM2. Black boxes indicate gene membership of cell-line-specific modules. Hierarchical clustering of genes from CODIM2 shows two major components (labeled in blue and green) and three smaller groups (dark green, brown, light blue). (**B**) Projection onto the STRING protein–protein network reveals a densely interconnected component (enriched in blue set) corresponding to fatty acid and (chole-)sterol metabolism (Supplementary Table 5 and Supplementary Figure 7). The second component (enriched in green set) is enriched in ER-related processes, such as protein N-linked glycosylation, transport and unfolded protein response. Genes subjected to functional knockdown assays of LDL uptake and free cellular cholesterol levels are highlighted in red or orange depending on whether validated as a functional regulator or not, respectively (two or more siRNAs having consistent effect with absolute $z$-score $>1$ and FDR-corrected $P$-value $<0.01$, Supplementary Table 6). (**C**, **D**) Representative image of fixed cells taken by automated widefield fluorescence microscopy. Cells have been transfected for 48 h with siRNA targeting indicated gene, and experiments of (**C**) LDL-uptake assay with fluorescently labeled DiI–LDL and (**D**) staining with cholesterol binding dye filipin was performed. Total intensity of fluorescent signal in segmented spots/area has been quantified as indicated with arrows.

(Szklarczyk *et al*, 2011)) showed that one of these sub-modules corresponds to cholesterol biosynthesis pathways (Supplementary Figure 7), whereas the other one forms a loosely connected network enriched in stress-related genes associated to the ER (Figure 3B). Recently, several lines of evidence have connected these processes and revealed the functional relevance of biological processes in the ER for the metabolism and transport of cholesterol (Colgan *et al*, 2007; Fu *et al*, 2012). Therefore, we examined whether genes in CODIM2 that are poorly characterized or have not previously been linked to this pathway could be novel regulators of cholesterol homeostasis. Twenty-four such genes were knocked down with small interfering RNA (siRNA) and their effects on LDL uptake (Ghosh *et al*, 1994) and/or free cholesterol levels (Börnig and Geyer, 1974) were experimentally assessed with quantitative microscopy-based assays (Bartz *et al*, 2009). These experiments confirmed that 10 out of 24 (42%) tested genes are indeed novel modulators of cellular cholesterol homeostasis; unexpected examples include FGFBP1 (fibroblast growth factor binding protein 1) and DUSP4 (dual specificity phosphatase 4) (Figure 3 and Supplementary Table 6). This validation rate exceeds that of a successful previous study, which used these assays to test candidates derived from targeted expression profiling and literature survey (Bartz *et al*, 2009). The experimental confirmation of as many as ten predictions highlights the power of our computational approach in defining functionally coherent transcriptional modules from large-scale drug perturbation experiments and demonstrated the utility of drug-induced modules for functional annotation of genes.

## Towards drug repositioning from drug-induced modules

Drug repositioning, that is the application of existing drugs to new indications, on the basis of computational prediction, has gained increasing attention (Campillos *et al*, 2008; Keiser *et al*, 2009; Iorio *et al*, 2010; Gottlieb *et al*, 2011). Our drug-induced transcriptional modules provide a rich source of novel leads for systematic drug repositioning. As a proof-of-concept, we suggest new chemotherapeutic agents by exploring the FDA-approved drugs in CODIM1, which is composed of two prototypical submodules characterized by inverse gene expression patterns (Figure 4A). One of them can be attributed to blocking cell cycle during G1/S transition (e.g., by methotrexate), whereas the other pattern corresponds to a G2/M arrest (e.g., by paclitaxel; Crawford and Piwnica-Worms, 2001; Whitfield *et al*, 2002). The vast majority of compounds found in CODIM1 are known cell cycle blockers, for instance, antifolate drugs (e.g., methotrexate and pyrimethamine), which block purine and pyrimidine synthesis, and thus DNA and RNA synthesis, and microtubule inhibitors (e.g., paclitaxel), which are used in anticancer therapy (Figure 4A). In addition to known cell cycle blockers, CODIM1 also comprises nine drugs that were not previously reported, to our knowledge, as cell cycle inhibitors in the literature and, hence, are candidates for drug repositioning. We selected three of them for further experimental validation: (i) vinburnine, a vasodilator; (ii) sulconazole, a topical antifungal; and (iii)

mephentermine, a cardiac stimulant. With cell viability assays, we examined whether these compounds indeed show an inhibitory effect on human cell lines. We did not observe reduced cell viability after mephentermine treatment (neither in HL60 nor in MCF7 cells), possibly because the transcriptional signature of mephentermine appeared weaker and more cell-line-specific than that of other drugs in CODIM1 that were well known to interfere with DNA replication (such as methotrexate, trifluridine and etoposide). However, for both sulconazole and vinburnine the predicted effect could be confirmed in HL60 and MCF7 with IC50 values of 6.1 μM (HL60) and 1.7 μM (HL60), respectively (Figure 4B and Supplementary Figure 8).

Moreover, the direct effects of sulconazole and vinburnine on the cell cycle were evaluated by propidium iodide (PI) staining and FACS analysis (Figure 4C and Supplementary Figure 9). Vinburnine treatment led to G2/M arrest in HL60 cells within 24 h, whereas prolonged treatments (48 and 72 h) enhanced the sub-G1 accumulation, indicating an apoptotic phenomenon (Figure 4C). Sulconazole, on the other hand, induced apoptosis (sub-G1 phase) of HL60 cell line within 24 h (Figure 4C), similar to the effects of the known apoptosis-inducing compound betulinic acid (Faujan *et al*, 2010). These experiments clearly confirm the inferred biological role of CODIM1 in cell cycle regulation and highlight the potential of its drug members as novel cell cycle blockers.

## Inferring context-dependent MOAs using cell-type-specific modules

Although conservation across multiple systems can increase the robustness of inferences, cell-type-specific responses to a drug could indicate a very selective effect, which necessitates cell-line-specific analysis. We thus tried to validate a few drugs that were unexpected in some of the system-specific modules and that might thus be candidates for drug repositioning or, at least, can serve as respective leads. To this end, we further investigated three cell-line-specific modules, each of which clearly showed enrichment for particular drug targets. Novel activity on the associated drug target was experimentally assessed for each of 10 repositioning candidates, which were selected to be structurally dissimilar to any drug contained in the respective module and known to interact with this target (Tanimoto similarity <0.5, see Materials and Methods). For example, the peroxisome proliferator-activated receptor sub-type-γ (PPARγ) is the main target of antidiabetic drugs (Vamecq and Latruffe, 1999) and is expressed in the PC3 cell line, but neither in HL60 nor in MCF7 cell line (Lamb *et al*, 2006). Our investigation among modules in PC3 revealed that the drug members of module PC3-9 were mainly comprised of PPARγ activators (Figure 5A and Supplementary Table 5). We therefore hypothesized that other drugs from PC3-9 also modulate PPARγ (Figure 5A). For one such candidate, zaprinast, an experimental phosphodiesterase inhibitor, target-binding assays confirmed that it is indeed a novel modulator of PPARγ ($K_i$: 14 μM), warranting further investigation of this drug in the context of diabetes (Figure 5D).

The estrogen receptor is known to be expressed specifically in MCF7 (Lamb *et al*, 2006) and is the therapeutic target for,

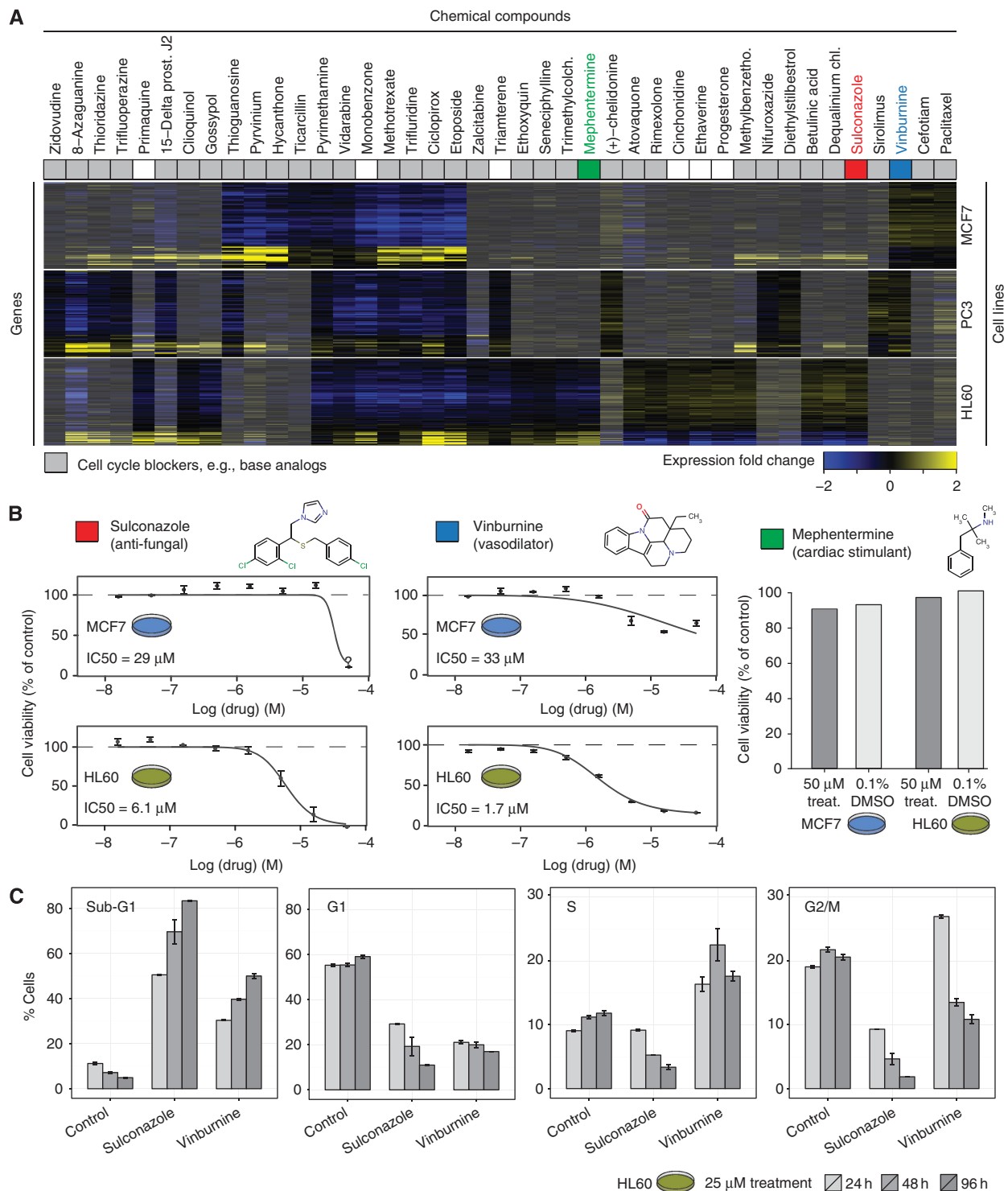

**Figure 4** Prediction of novel cell cycle inhibitors based on CODIM1. (**A**) Heatmap: expression fold change in each cell line with shaded columns, indicating chemical treatments not present in the respective cell-line-specific module. Grey: compounds with a known effect on cell cycle (Supplementary Table 5). Although, six well-known cell cycle blockers were detected in all cell lines delineating a module core in CODIM1, 26 additional cell cycle blockers were only detected as part of this module in one or two of the three cell lines (shaded columns). This variability highlights the benefits of aggregating information across cell lines. (**B**) Three chemicals not previously known to inhibit cell cycle were examined in cell viability and proliferation assays. For sulconazole and vinburnine dose-dependent reduction of cell viability and proliferation were confirmed in two cell lines with IC50 values as indicated. For mephentermine, an effect on cell viability and proliferation could not be detected in either cell line (treat., treatment). (**C**) Control and drug-treated HL60 cells were attributed to different cell cycle phases according to their DNA content (PI, FACS analysis) for three treatment durations (24, 48 and 96 h). The error bars represent the s.e.m. Sulconazole (25 μM) treatment led to a marked increase in sub-G1 population across all time points (24, 48 and 96 h), which is indicative of an apoptotic cell population. Vinburnine (25 μM) induced a G2/M arrest within 24 h similar to Nocodazole (Supplementary Figure 9) and further treatment (48 and 96 h) resulted in increased apoptosis in HL60 cells.

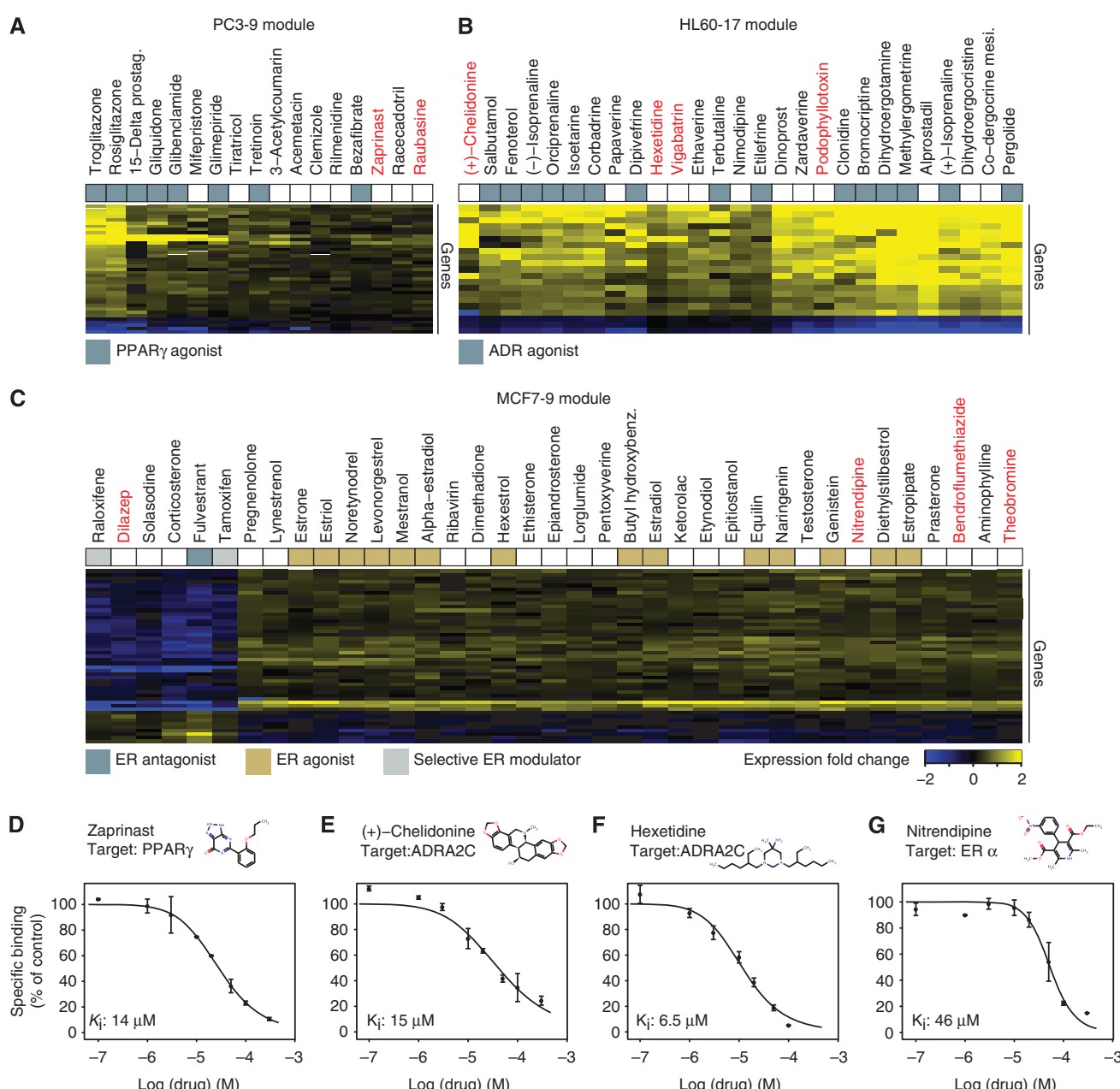

**Figure 5** Drug repositioning from drug-induced transcriptional modules. Three examples of cell-line-specific drug-induced modules that were enriched for pharmacological classes of (**A**) peroxisome proliferator-activated receptor activators (PC3-9), (**B**) α-adrenergic 2 agonists (HL60-17), (**C**) estrogen receptor-α modulators (MCF7-9; see Supplementary Tables 4 and 5 for a complete list of modules). For each module (as detected by the biclustering approach), a heatmap (**A**–**C**) was drawn to illustrate gene expression changes (fold change) under various drug treatments. Drugs in modules were characterized with respect to their molecular targets. Specific action on the main target associated with the respective module **A**–**C** is indicated by colored boxes. Novel drug–target relationships were inferred for drug members not previously known to modulate the main targets associated with these modules. Ten predicted modulators were experimentally tested (red labeled drugs in **A**–**C**). (**D**–**G**) Four of these predictions could by verified with *in vitro* binding assays (bold face). In three cases $K_i$ values lower than 15 μM confirmed strong binding, whereas ERα affinity of nitrendipine (46 μM) was considered ambiguous (Lounkine *et al*, 2012; Supplementary Figure 10).

e.g., breast cancer treatment and osteoporosis. To infer novel modulators of the estrogen receptor, we investigated drug members of module MCF7-9 that is enriched for estrogen receptor agonists and antagonists (Figure 5C and Supplementary Table 5). We tested and confirmed that nitrendipine, a dihydropyridine calcium channel blocker, weakly binds to the estrogen receptor ($K_i$ about 46 μM, Figure 5G).

We further analyzed the HL60-17 module in which drugs with adrenergic activity were overrepresented (Figure 5B). Adrenergic agonists have therapeutic effects in various disorders, including vasoconstriction and attention deficit-hyperactivity disorder. Novel candidates in this module not previously described to have any adrenergic activity, were subjected to target-binding assays for ADRA2C. We confirmed the activity for both hexetidine (antibacterial, $K_i$: 6.5 μM

and $(+)$-chelidonine, $K_i$: 15 μM; Figure 5E, F). In total, we successfully validated target-binding activity for 4 out of 10 candidate drugs selected from these modules (Supplementary Figure 10). Although the binding affinities measured may be too low for immediate pharmaceutical application, the respective drugs can at least be considered as novel leads, which can be further optimized towards novel therapeutic uses (Wermuth, 2004; Chong and Sullivan, 2007) and, thus, corroborate the value of our approach for drug repositioning.

## Conclusions

We identified and annotated a large set of mammalian drug-induced transcriptional modules. We analyzed the modules in a cell-type-specific manner, which not only allowed us to uncover cell-type-specific drug responses, but also to derive a lower limit for the conservation of drug-induced transcriptional modules across cell lines and organisms. In contrast to previous studies, which mainly focused on drug–drug relations based on similarity of whole-expression profiles (Iorio *et al*, 2009, 2010; Perlman *et al*, 2011), we also characterized the biological roles of the genes responding to drug perturbations. The biological relevance of transcriptional modules was underlined by the experimental validation of 10 genes as novel modulators of cholesterol homeostasis. Moreover, we discover novel MOAs for six drugs (two in conserved and four in cell-line-specific drug-induced transcriptional modules). These results provide initial evidence for the potential of the transcriptional modules in drug repositioning and highlight the value of our approach for improving our understanding of drug action.

With more fine-grained, large-scale chemogenomics data sets becoming publicly available (e.g., Barretina *et al*, 2012; Garnett *et al*, 2012), the presented framework should become much more powerful, allowing a more accurate delineation of transcriptional modules. We expect that the estimate of the extent to which transcriptional responses are conserved between cell types and tissues may considerably increase due to higher resolution. Moreover, drug-induced transcriptional modules that are significantly associated with specific (off-) targets or side effects could be utilized as transcriptional markers (Afshari *et al*, 2011; Iorio *et al*, 2012) to systematically evaluate the efficacy and safety of new chemicals in early stages of the drug development process. We believe that better profiling and characterization of the responses in different cell types, in turn, will lead to a more systematic understanding of pharmacological and toxicological properties of chemicals.

## Materials and methods

### Microarray data source and preprocessing

The CMap (build 02, http://www.broadinstitute.org/cmap/) contains 6100 genome-wide expression profiles comprising the responses of four human cancer cell lines to treatment with 1309 small molecules (hereafter will be referred as 'drugs'; Lamb *et al*, 2006). Filtering and normalization of this data set was performed as reported previously (Iskar *et al*, 2010), in order to obtain a comparable set of experiments in each of the three major cell lines in CMap (HL60, human promyelocytic leukemia cell line; MCF7, human breast adenocarcinoma cell line; PC3, human prostate cancer cell line) and to reduce batch effects by adjusting expression changes against a batch-specific background. In cases where multiple replicate experiments for the same drug in the same cell line were present, we selected one representative per cell line, such that consistency across cell lines is maximized based on Pearson's correlation, and discarded the other replicates. Further analysis was restricted to genes (probe sets) expressed in at least one of the cancer cell lines. A probe set was considered expressed if the 'present' call ratio exceeds 10% of experiments in any of the cell lines (with gene- and experiment-wise 'present' calls made as proposed earlier (Liu *et al*, 2002)). In the second step of probe set removal, we chose a single probe set per gene at random to avoid analysis bias towards genes represented by multiple probe sets. As a result, expression response of 8964 distinct genes from three human cancer cell lines to treatments with the same set of 990 distinct drugs were retained for this study.

In addition, a large-scale data set of gene expression changes in rat liver of compound-treated rats were included in this study (DrugMatrix, GEO accession number GSE88583; Natsoulis *et al*, 2008). It contains 5312 microarray experiments for multiple concentrations of 344 compounds (and diet restriction) profiled at different time points after administration (in most instances 6 h and 1, 3 and 5 days) with biological replicates. Microarray data was normalized and summarized into drug-induced expression profiles in multiple steps similarly as described (Huang *et al*, 2009c). Analysis was restricted to probe sets that are in common between all microarray platforms used (GE Healthcare/Amersham Biosciences CodeLink UniSet Rat I Bioarray, EXP5280X2-584, EXP5280X2-613 and EXP5280X2-648). Subsequently, to facilitate interexperimental comparisons, all experiments were subjected to quantile normalization (Bolstad *et al*, 2003). Probe sets with more than 10% missing values were excluded and the remaining missing values were filled in by mean imputation, which is replacement with the probe set-specific mean expression value. For each gene, we only retained the single probe set with the highest variance. Subsequently, gene expression values were first converted to fold changes by subtracting the mean value of the corresponding controls and then divided by the s.d. (added with 0.1 quantile of all s.d.) across all samples to obtain expression $z$-scores relative to background. These were subsequently averaged across replicates. Lastly, we established an orthology relation between rat and human genes based on NCBI Homologene (Sayers *et al*, 2011) and retained only those probe sets with a one-to-one ortholog among the genes represented in the processed CMap data set (see above). This procedure retained 1743 drug-induced gene expression profiles of rat liver tissue for 3618 genes in response to 344 compounds in multiple doses and time points (for simplicity, different doses and time points for the same drug were treated as independent experiments in the subsequent module identification, Supplementary Figure 1).

### Identifying transcriptional modules

A transcriptional module is defined as a set of genes that are coherently changing under a subset of conditions. To identify such modules, we used an unsupervised biclustering approach, called the Iterative Signature Algorithm (Bergmann *et al*, 2003; Ihmels *et al*, 2004; as implemented in the R packages, 'isa2' (version 0.3) and 'eisa' (version 1; Csárdi *et al*, 2010) belonging to the R Bioconductor framework (Ihaka and Gentleman, 1996; Gentleman *et al*, 2004)). In this study the four matrices of drug-induced expression $z$-scores (three human cell lines and rat liver, preprocessed as described above) were analyzed separately using the same ISA workflow.

As described in previous studies (Kutalik *et al*, 2008; Csárdi *et al*, 2010; Henrichsen *et al*, 2011), ISA proceeds in several steps. First, transcriptional modules are detected for a wide variety of different threshold values imposing a minimum on the number of included genes (threshold varied from 5 to 2 in decrements of 0.2) and conditions, i.e., drug treatments (threshold varied from 4 to 1 (2 in rat liver) in decrements of 0.2); the ISA algorithm performed 20 000 (internal) re-runs with different random starting points for each threshold. The resulting outputs are highly redundant (see Supplementary Figure 2 for an exploration of redundancy among and sensitivity to different parameter sets) and therefore require additional redundancy removal steps (Csárdi *et al*, 2010). Here we customized the standard ISA procedures as follows, First, we enriched for drug-induced transcriptional modules over constitutive modules

that were also apparent from untreated background experiments. Although carried out on drug-treatment experiments only, the modules identified by ISA may contain generic coexpression modules that correspond to intrinsic, constitutive cellular processes rather than specific responses to drug treatments. We therefore removed all ISA biclusters if at least 10% of gene pairs within the module showed strong coexpression in untreated samples as well (as defined by a Pearson's correlation $>0.6$, see Supplementary Table 2 for robustness analysis with respect to the actual parameters used). Second, we discarded very small, likely spurious modules and retained only biclusters with a minimum of 20 genes and 5 drugs (10 for rat liver). Finally, we applied the standard ISA redundancy removal to reduce the number of very similar biclusters that are the result of many randomly initialized runs converging to highly similar result sets (Csárdi et al, 2010). Before this removal step, all transcriptional modules were sorted by gene and drug thresholds aiming to preferentially retain medium-sized modules (first prioritizing gene threshold values from 4 to 3 over ones from 5 to 3.2 and finally 2.8 to 2 for cell line data, and prioritizing values from 3 to 2 over ones from 5 to 3.2 for rat liver data; second, prioritizing drug threshold values from 3 to 2 over 4 to 3.2 in both cell line and rat liver data followed by values from 1.8 to 1 in cell line data). On the basis of this prioritization, redundant modules were filtered in sequential order following the developers' recommendations (Csárdi et al, 2010), using a correlation threshold of 0.3 to determine redundant biclusters (see Supplementary Table 1 for a parameter robustness analysis). Lastly, in addition to ISA redundancy removal, modules from the same cell line showing significant gene overlap with a module of higher priority (hypergeometric test, $P$-value $<1E-5$) were discarded sequentially to further reduce redundancy in terms of gene overlap (see Supplementary Table 1 for a parameter robustness analysis).

## Comparison of transcriptional modules between data sets

Drug-induced transcriptional modules were compared with each other using a hypergeometric test to assess significance of overlap between gene members (similarly done in Oldham et al (2008)). With a reciprocal best-hit approach we identified associations—CODIMs—between transcriptional modules across cell lines and species (with $P$-value $<0.01$ after a false discovery rate (FDR) correction for multiple hypothesis testing; Figure 2B and Supplementary Figure 3). In addition, we tested all associations between modules for overlap between drugs inducing these modules (but not for rat liver, as the number of drugs in common with this data set was too low to provide a solid statistical basis for comparison of drug overlap). Significant overlaps between drug sets (Fisher's exact test, FDR-corrected $P$-value $<0.01$) were indicated in Figure 2B, Supplementary Figure 3 and Supplementary Table 3. Finally, CODIMs were obtained as the union of gene and drug members of the corresponding transcriptional modules from human cancer cell lines.

## Assessing functional coherence of transcriptional modules

In order to assess whether transcriptional modules were enriched for functionally associated gene pairs, we compared modules from human cancer cell lines and rat liver to the comprehensive STRING database of protein–protein associations for *Homo sapiens* and *Rattus norvegicus*, respectively (Szklarczyk et al, 2011). In total, 126 622 reliable associations between 6556 human proteins were extracted from the STRING network considering only the experimental and database evidence with intermediate to high confidence (the default STRING evidence scores $>0.4$), while excluding methodologically analogous coexpression associations (von Mering et al, 2005). Similarly, 11 988 reliable associations between 1869 rat proteins were retrieved from STRING having a combined evidence score over 0.4 and again excluding analogous coexpression associations. For each module, we calculated the proportion of functionally associated gene pairs (excluding self-relations) within the module and compared this against the distribution obtained for the proportion of functionally associated pairs in 1000 random gene sets of matched sizes.

## Characterization of drug-induced transcriptional modules

We characterized both individual transcriptional modules along with their unified CODIMs in detail by mining information for both gene and drug members using several annotation resources. We used the Database for Annotation, Visualization and Integrated Discovery (DAVID knowledgebase; Huang et al, 2009a, 2009b) to test for enriched gene function and to identify biological themes among these. For each module, the DAVID web service provided a ranked list (enrichment score $>1.3$) of functionally relevant annotation clusters that represent a summary of several annotation categories, including GO (Ashburner et al, 2000), KEGG (Kanehisa et al, 2012) and BioCarta pathways (Nishimura, 2001). In this analysis, gene reference background was set to all genes for CMap, and rat liver data set provided as an input to the ISA algorithm.

The analysis of gene functions was complemented with a second approach, exploiting drug annotation resources for module characterization. For this, we extracted the ATC classification code (Andersen and Hvidberg, 1981; Pahor et al, 1994) for 677 approved drugs from the set of chemicals in CMap and 4331 chemical–protein interactions for 493 CMap compounds from the STITCH database (Kuhn et al, 2011), considering only reliable database and experimental evidence with high confidence (any of the two scores $>0.7$). In addition, 23 458 associations between 1106 side effects and 373 drugs contained in CMap were obtained from the SIDER database (Kuhn et al, 2010). Lastly, a library of chemical fragments ($>6$ atoms) generated by exhaustive molecular fragmentation of 892 CMap drugs (93%) ($<41$ bonds) was queried against chemical structures of 989 drugs from CMap using substructure search (Steinbeck et al, 2003; Guha, 2007). Chemical structural similarity was assessed with 2D Tanimoto coefficients based on hashed fingerprints with a default length of 1024 bits (default path length 8) calculated using the CDK (Willett et al, 1998; Martin et al, 2002; Steinbeck et al, 2006). Drug annotation terms linked with at least five drugs were retained for further analysis. In total, enrichment of annotation terms of 160 drug targets, 56 ATC class (second level), 571 side effects and 813 chemical fragments were tested for all modules from cell lines, including CODIMs, using Fisher's exact test with FDR correction (within each data set using a $P$-value cutoff of $<0.1$, except for chemical fragments with a $P$-value $<0.01$; Supplementary Table 5).

## Cell viability assays and cell cycle analysis

HL60 (CPQ-054) was cultured as suspension cells in RPMI-1640 containing 10% FCS and penicillin/streptomycin. MCF7 (CPQ-072) cells were cultured as adherent cells in DMEM containing 10% FCS and penicillin/streptomycin. Cells were cultured at 37 °C in an atmosphere of 5% (HL60) or 10% (MCF7) $CO_2$. For the assays, cells (5000 cells/ well) were seeded in 150 µl medium on a 96-well cell culture plate and incubated overnight at 37 °C. The next day, cells were treated with different concentrations of selected chemicals from CMap. After 72 h, the viability of treated cells was measured after incubation for 4 h with the indicator dye Alamar Blue and fluorescence measurement at 590 nm. The single concentration of mephentermine (50 µM) was tested in quadruplicates. The IC50 curves for the predicted compounds (sulconazole, vinburnine) and their corresponding negative controls (butoconazole, moxisylyte hydrochloride) were prepared in eight semilog dilutions starting at 1E-5M and tested in duplicates. The reference compound staurosporine was prepared in eight decalog dilutions starting at 1E-5M. Standard solvent controls were performed using 0.1% DMSO. Treatment of cells with 0.1% DMSO and 1E-5M staurosporine served as high control (100% viability) and low control (0% viability), respectively. Raw data were converted into percent viability relative to high controls (solvent 0.1% DMSO) and low controls (1E-05M staurosporine), which were set to 100 and 0%, respectively. IC50 values were calculated using R package 'drc' (Ritz and Streibig, 2005) by fitting dose response data to a two-parametric log-logistic function with 0% growth as bottom constraint and 100% growth as top constraint. For cell cycle analysis, HL60 cells (ATCC CCL-240) were grown for 6 h with an initial concentration of 0.4 Mio/ml in 3 ml of medium before sulconazole (25 µM) and vinburnine (25 µM)

(nocodazole, 200ng/ml as reference) were added. For three time points (24, 48 and 72 h), (un)treated cells were collected, fixed in 80% methanol for 24 h and further rehydrated in PBS/1% FCS at room temperature for 30 min before RNAse A (100 μg/ml) and PI (50 μg/ml) treatment. FACS (FACSCalibur, BD Biosciences) was employed to determine cell cycle distribution based on ~5000 events. The Fluorescence detector FL3 was calibrated in a way that the first (FL3-A) peak of untreated sample was set around 200 units representing G1 population. For each sample, the ratio of cells in each cell cycle stage were determined based on their DNA content using 'Flowing software' (Supplementary Figure 9) at ~200 units defined as G1, below G1 peak (~200) as apoptotic cells (sub-G1), ~400 units as G2/M cells and those between G1 and G2 peak was considered as S phase. Events with over 400 units was defined as endo-reduplicated cells.

## *In vitro* binding assays

*In vitro* binding assays were performed by the company Cerep for α-2 adrenergic receptor (agonist, ref. 3443), estrogen receptor-α(ERα, agonist, ref. 0484) and PPARγ (agonist, ref. 0641). Similarly, Cerep carried out the cellular assays of 2C-1 adrenergic receptor (α1B, agonist effect, ref. 1901) and ERα (antagonistic effect, ref. 3495). All chemicals were purchased from Sigma and were tested in the following assays: Zaprinast (Z0878) and raubasine (41111) in PPARγ; dilazep (D5294) in ERα antagonistic assay; bendroflumethiazide (B5775), nitrendipine (N144) and theobromine (T4500) in ERα-binding assay (agonist); and hexetidine (259187), vigabatrin (V8261), podophyllotoxin (P4405) and (+)-chelidonine (54274) in α2B and α1B assays. The effect of chemicals were initially evaluated at 50 μM, and $K_i$ values were further determined for chemicals with more than 40% activity in initial assessment. As explained above, R package 'drc' (Ritz and Streibig, 2005) was utilized to calculate IC50 values using a two-parametric log-logistic function with 0% as minimum and 100% as maximum. $K_i$ values were generated from the IC50 values using Cheng–Prusoff equation. Three predictions were confirmed as hits with $K_i$ values lower than 15 μM and the binding activity of nitrendipine was considered ambiguous as being between 25 and 50% at 30 μM, while the rest of the predictions were labeled as disproved (Lounkine *et al*, 2012).

## Functional cholesterol assays using siRNA knockdowns

Knockdown was performed by pre-designed 21 nt Silencer Select siRNAs or 25 nt Stealth RNAi™ siRNAs from Life Technologies. Assays for LDL uptake and free cholesterol (Filipin assay) and image acquisition was performed as described in Bartz *et al* (2009). In short, for both assays Hela Kyoto cells were transfected with siRNA for 48 h. For the LDL-uptake assay, cells were starved in FCS-deficient medium overnight and 1% (w/v) HPCD 45 min before adding DiI-labeled (3,3′, dioctadecylindocarbocyanine) LDL for 20 min. For the Filipin assay, cells were fixed 48 h post transfection and stained with fluorescent dye filipin binding to free cholesterol. Fixed cells were imaged on an Olympus ScanR system. All images were manually quality controlled in order to exclude out-of-focus images and otherwise not analyzable images (e.g., too many cells, dust particles etc.). Automated image analysis was performed using the Open Source software Cellprofiler. An additional module (MorphoQuant) was programmed to specifically detect dots using convolution with a mask previous to thresholding. Structures were detected by thresholding above background in the cell. For these structures the total intensity was quantified. For the LDL-uptake assay, column bias within plates was reduced by subtracting column-wise estimated background based on controls using local polynomial regression. The effect of siRNA knockdowns versus controls were evaluated using linear mixed-effect models (as implemented in the R package 'nlme' (version 3.1-103)), with siRNA treatment as the fixed factor and plate as the random factor. For each assay, *P*-values were adjusted by FDR correction. Experimentally tested genes were considered as positive hits if two or more of its gene-specific siRNAs had consistent and significant effect(s) on either LDL uptake and/or free cholesterol (Filipin assay) in comparison with controls (using a cutoff of absolute *z*-score >1 and FDR-corrected *P*-value <0.01; Supplementary Table 6).

## Data availability

Supplementary Data such as drug-induced modules and CODIMs are available from http://codim.embl.de in human and machine readable formats. The web resource also provides functionality for drug and gene search.

## Supplementary information

## Acknowledgements

We thank John P Overington, Sevi Durdu, Christina Besir, Elisabeth Georgii, Christian Tischer and members of the Bork group for helpful discussions.

*Author contributions:* MI, GZ and MC performed data analysis. MI implemented analysis pipeline. P Blattmann performed cholesterol assays. Cell viability assays were done at ProQinase GmbH (Freiburg, Germany) and target-binding assays were done at Cerep, SA (Poitiers, France), both with help from ACG. P Bork, MI, GZ, MC, VN, ACG, HR, KHK, RP and MK designed the study and/or advised on the analyses. MI, GZ, VN and P Bork wrote the manuscript with contributions from all other authors.

## Conflict of interest

The authors declare that they have no conflict of interest.

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
