## [Review Process File · Molecular Systems Biology]

Characterization of drug-induced transcriptional modules: towards drug repositioning and functional understanding

Murat Iskar, Georg Zeller, Peter Blattmann, Monica Campillos, Michael Kuhn, Katarzyna H. Kaminska, Heiko Runz, Anne-Claude Gavin, Rainer Pepperkok, Vera van Noort, Peer Bork

Corresponding author: Peer Bork, EMBL

Review timeline:

Submission date:	21 January 2013
Editorial Decision:	28 February 2013
Revision received:	26 March 2013
Accepted:	28 March 2013

Editor: Thomas Lemberger

Transaction Report:

1st Editorial Decision

28 February 2013

Thank you again for submitting your work to Molecular Systems Biology. We have now heard back from the three referees who accepted to evaluate the study. As you will see, the referees find the topic of your study of potential interest and are supportive. They raise however a series of concerns and make suggestions for modifications, which we would ask you to carefully address in a revision of the present work. The recommendations provided by the reviewers are very clear in this regard. Thank you for submitting this paper to Molecular Systems Biology.

REFeree REPORTS:

Reviewer #1 (Remarks to the Author):

It was a great pleasure to read the manuscript by Iskar et al. I have to imagine that I am seeing this manuscript after a revision, because it is in very good form. The approach is novel and superb in its integration of several disparate data sets, and the experimental validations add strong support for the validity of the approach. Although most reviewers feel they must manufacture problems with papers to seem like they are doing their job, I am happy to say that I don't see any major issues with this paper, and I think it would be of high interest to the MSB readership. If only all other journals would send me papers that are this interesting and well-written. The only minor issues I see are as follows:

- I would like to see some discussion as to possible reasons why mephentermine did not validate. I applaud the authors for including a potential false positive in the results, but I think this failed validation was interesting, and would like to hear a bit more about the possible biology behind this.

If you look at Fig 4A, you definitely see 2-3 classes of transcription patterns, where Mephentermine seems to be the edge of one class, so wondering if there is any interesting biology to glean from this finding.

- The authors do not indicate if they will make the data publicly available. Indeed a public CODIM resource would provide great value to the community as it is a rich source of information that suggest likely more hypotheses than the investigators themselves could chase down. The corresponding author has been generous in making public resources for this type of systematic data in the past, so I would strongly encourage publishing the module data as a public resource along with manuscript publication.

Reviewer #2 (Remarks to the Author):

General comments:

The authors identified a set of drug-induced transcriptional modules by applying a bi-clustering method to gene expression data. It is an interesting approach toward gene function prediction and drug repositioning, and experimental validation is appreciated. One concern is the robustness of the extracted modules.

Specific comments:

1. In the process of identifying transcription modules, the authors performed many filtering procedures for the redundant modules produced by the ISA algorithm. I think that the resulting modules depend heavily on various parameters in the ISM algorithm and filtering thresholds (e.g., ratio of gene pairs in page 17, Pearson correlation in page 17, correlation threshold in page 18). Why did the authors determine such parameters and thresholds in this analysis? Is it possible to determine appropriate parameters objectively? Even if different parameters are used, the result with the same argument can be obtained?
2. The authors evaluated chemical structural similarity drugs by Tanimoto similarity in the "Inferring context-dependent MOAs" section. Which fingerprints did they use? The significance of 0.5 depends on the choice of chemical fingerprints.
3. The authors confirmed the prediction by the in vitro binding assays in the experimental validation process. However, is 15 nM enough for pharmaceutical purpose?
4. The authors assessed the modules by using the STRING network with reliable associations. What does it mean by "score of 400"? It is not understandable.
5. What does it mean by "q-value" in Figure 2? Should it be "p-value"?
6. Figure 5 shows 3 modules: PC3-9 Module, HL60-17 Module, and MCF7-9 Module. How did the authors name the modules? Does each module correspond to one of 23 CODIMs?

Reviewer #3 (Remarks to the Author):

In this paper, the authors applied bi-clustering strategy to identify drug-response gene expression modules across multiple cell lines and in vivo rat model. The authors further demonstrated that the drug-expression file modules can be applied to revealing drug mode of actions, and providing new opportunities for drug repurposing. The results are interesting and impressive. It may have big impact on the field of drug discovery.

Strengths of the paper

1. Novel use of bi-clustering algorithm.
2. The use of multiple cell lines and in vivo models to identify common bi-clusters.

3. The detailed functional annotations of proposed modules
4. The extensive experimental validations of predictions.

Issues need to be addressed:

1. Compared with other conventional methods for statistical based clustering of gene expression profile, or biological driven protein-protein interaction network-based clustering, what is the performance gain of the bi-clustering strategy? What is the sensitivity and specificity of the method?
2. Authors should discuss the limitations of the bi-clustering strategy. For example, bi-clustering clusters drugs and genes in disjointed set. It may miss an important aspect of drug actions, i.e drugs may modulate multiple pathways. One example is cox-2 inhibitors that impact both cox-2 dependent and independent pathways.
3. Majority of experimentally valid bindings are quite weak. It implies either these chemicals are promiscuity binders or the predicted off-target has little impact on the phenotype than its primary target. If it is the first case, there are many undetected false negatives. For the second case, it makes more sense that the drug is clustered with its high-affinity primary binders. One of such cases is zaprinast whose primary target is PDE. If the clustering does not detect the primary target, I think it should be considered as a false positive. The use of this drug for repurposing may be unjustified, as it may induce dominant phenotype response through its much more strongly bound primary target.
4. Genetic circuit is dynamic and context-dependent. The definition of conserved gene/drug modules only makes sense when the primary drug modulation pathway(s) have not been significantly rewired. Thus, it may important to take the background expression profiles into account when compare the modules derived from their differential changes. More cautions should be taken when comparing in vitro modules and in vivo modules. In addition to the issue of network rewiring, pharmacokinetics may play critical roles in drug action in vivo. Moreover, dramatically different from in vitro drug binding, the drugs are rarely in an equilibrium state in vivo. Thus the determinant factor of drug-target interaction in vivo is drug resident time, instead of binding affinity measured by IC50 or Ki. I am not convinced by authors argument that drug induced gene modules are extensively conserved across cell lines, species, and in vitro and in vivo conditions.

Other issues:

1. During the bi-clustering stage, there are a number of pre-defined threshold. What is the rationale in choosing these thresholds, and how sensitive of the bi-clusters to these thresholds?
2. It is not clear how to process the time-course gene expression data of rat liver model. Although the authors provide a reference, a brief summary may be useful for readers.

1st Revision - authors' response

26 March 2013

Detailed point by point response to the reviewer's comments:

The reviewers' comments are in *italics*, followed by our comments as well as references to revisions in the manuscript.

Reviewer #1 (Remarks to the Author):

"It was a great pleasure to read the manuscript by Iskar et al. I have to imagine that I am seeing this manuscript after a revision, because it is in very good form. The approach is novel and superb in its integration of several disparate data sets, and the experimental validations add strong support for the validity of the approach. Although most reviewers feel they must manufacture problems with papers to seem like they are doing their job, I am happy to say that I don't see any major issues with this paper, and I think it would be of high interest to the MSB readership. If only all other journals

would send me papers that are this interesting and well-written. The only minor issues I see are as follows:

- I would like to see some discussion as to possible reasons why mephentermine did not validate. I applaud the authors for including a potential false positive in the results, but I think this failed validation was interesting, and would like to hear a bit more about the possible biology behind this. If you look at Fig 4A, you definitely see 2-3 classes of transcription patterns, where Mephentermine seems to be the edge of one class, so wondering if there is any interesting biology to glean from this finding.”

As the reviewer, we were intrigued by the two inverse transcription patterns apparent in CODIM1, one of which corresponds to cell-cycle arrest in G2/M (as confirmed by FACS-based cell-cycle analysis for vinburnine and known for paclitaxel). The other pattern is likely a signature of G1/S arrest (as inferred by the inducing drugs that are anti-folate compounds, topoisomerase inhibitors, and nucleoside analogs including methotrexate, trifluridine, etoposide, vidarabine, pyrimethamine, and 8-azaguanine, all of which are known to interfere with DNA replication). Although their transcriptional signature generally appears more robust across cell lines (compared to the former pattern that is more specific to HL60 and MCF7), we unfortunately failed to show an effect of mephentermine on cell viability. In the revised manuscript we added that this might be due to weaker and HL60-specific expression changes induced by this compound (maybe causing a spurious association in the biclustering and thus the unsuccessful validation).

Action taken: Added a potential explanation for this failed validation experiment to the manuscript (page 11 lines 275-279).

- “The authors do not indicate if they will make the data publicly available. Indeed a public CODIM resource would provide great value to the community as it is a rich source of information that suggest likely more hypotheses than the investigators themselves could chase down. The corresponding author has been generous in making public resources for this type of systematic data in the past, so I would strongly encourage publishing the module data as a public resource along with manuscript publication.”

To address this valid point, we have set up a supplementary web resource (<http://www.bork.embl.de/Docu/codim>) that provides access to the drug-induced transcriptional modules, their gene and drug members in both, human-readable and machine-readable formats. This web resource is also referred to in the revised manuscript. We plan to develop it further and integrate it with other web-based resources on drugs and their effects on biological systems.

Action taken: Included a section on Data Availability (page 25, lines 604-607) mentioning the new web resource.

Reviewer #2 (Remarks to the Author):

General comments:

“The authors identified a set of drug-induced transcriptional modules by applying a biclustering method to gene expression data. It is an interesting approach toward gene function prediction and drug repositioning, and experimental validation is appreciated. One concern is the robustness of the extracted modules.

Specific comments:

1. In the process of identifying transcription modules, the authors performed many filtering procedures for the redundant modules produced by the ISA algorithm. I think that the resulting modules depend heavily on various parameters in the ISM algorithm and filtering thresholds (e.g., ratio of gene pairs in page 17, Pearson correlation in page 17, correlation threshold in page 18). Why did the authors determine such parameters and thresholds in this analysis? Is it possible to determine appropriate parameters objectively? Even if different parameters are used, the result with the same argument can be obtained?”

We are thankful for this valid comment (which was similarly also made by reviewer #3) on the superficial presentation of robustness checks in the submitted manuscript. We did perform a parameter exploration and have now revised and extended the manuscript to include supplemental figures and tables showing the results of a thorough sensitivity analysis of modules with respect to the choice of ISA parameters. Most importantly, the modules presented in the manuscript were the result of a comprehensive exploration of the ISA parameter space followed by a redundancy removal step (similar to the recommendations of the ISA developers). A newly included supplemental figure (Suppl. Fig. 2) now shows that each of the final modules could indeed be detected with a wide range of ISA parameter settings, most of the modules even with >100 of the 256 parameter sets explored. Moreover we now also show the robustness of the final module set to the choice of the correlation and overlap cutoffs used for redundancy removal (Suppl. Table 1) and to the correlation threshold used to remove modules already detectable in untreated controls and thus not necessarily induced by drug treatments (Suppl. Table 2). Overall, we believe that these results convincingly show that the actual parameters chosen have little influence on the final module set described in the manuscript.

Action taken: Conducted parameter robustness analysis (new Suppl. Fig 2 and Suppl. Tables 1 and 2) and included necessary pointers in the manuscript (page 5, line 127).

2. *“The authors evaluated chemical structural similarity drugs by Tanimoto similarity in the “Inferring context-dependent MOAs” section. Which fingerprints did they use? The significance of 0.5 depends on the choice of chemical fingerprints.”*

We apologize for this omission. The Chemistry Development Kit was used to generate the standard (CDK) hashed fingerprints with a default length of 1024 bits (default path length 8). Based on these, Tanimoto (2D) similarity was calculated between structural fingerprints.

Action taken: Revised Methods section to clarify this point (page 21 and lines 511-514, pointer included on page 12, line 303).

3. *“The authors confirmed the prediction by the in vitro binding assays in the experimental validation process. However, is 15 μ M enough for pharmaceutical purpose?”*

The reviewer’s concern is valid that the experimentally determined affinities may be too weak to conclude that these drugs would have pharmaceutical potency against the novel targets without modifications (see also similar concern by reviewer #3 and our answer).

Action taken: Adjusted the discussion of experimental results to more carefully state that the confirmed prediction provide new leads against these targets, but may need further optimization (pages 12, 13, 14 and lines 297-298, 311-312, 328-332, included additional references).

4. *“The authors assessed the modules by using the STRING network with reliable associations. What does it mean by “score of 400”? It is not understandable.”*

This cutoff was indeed not motivated in the manuscript. We revised it to convert the scoring scheme used in the downloadable flat files to the one displayed in the STRING web interface. The flat files contain scores that are actually multiplied by 1000, hence such a score of >400 equals a score >0.4 in the online version. This default cutoff corresponds to associations with an intermediate to high confidence level. In STRING such a score means that in a benchmark set 40% of the interactions correspond to protein pairs that function in the same pathway.

Action taken: Modified Methods section (page 20, lines 479-480, 482) to clarify that this is a STRING default setting with the above meaning.

5. *“What does it mean by “q-value” in Figure 2? Should it be “p-value”?”*

We have modified Fig. 2, Methods and Supplementary Materials, replacing statistical jargon (q-value) by “FDR-corrected p-value” and clarified this in the revised main text.

Action taken: Necessary modifications were made throughout the manuscript (including captions of both Fig. 2 & 3; pages 19, 25, 33, 34 lines 463, 468, 602, 894, 917 and Supplementary Material).

6. *“Figure 5 shows 3 modules: PC3-9 Module, HL60-17 Module, and MCF7-9 Module. How did the authors name the modules? Does each module correspond to one of 23 CODIMs?”*

These are identifiers for modules detected in individual cell lines as indicated by the cell line prefixes (PC3, HL60 and MCF7) followed by a number in the order in which they were generated by the ISA procedure.

Action taken: Revised caption of Fig. 5 (page 35, lines 948-949) to include pointers to Suppl. Fig. 3, which shows all modules and to which CODIMs they belong, as well as to Suppl. Table 4, which summarizes how these were annotated. Full information is provided in Suppl. Data Set 1.

Reviewer #3 (Remarks to the Author):

“In this paper, the authors applied bi-clustering strategy to identify drug-response gene expression modules across multiple cell lines and in vivo rat model. The authors further demonstrated that the drug-expression file modules can be applied to revealing drug mode of actions, and providing new opportunities for drug repurposing. The results are interesting and impressive. It may have big impact on the field of drug discovery.

Strengths of the paper

1. Novel use of bi-clustering algorithm.
2. The use of multiple cell lines and in vivo models to identify common bi-clusters.
3. The detailed functional annotations of proposed modules
4. The extensive experimental validations of predictions.

Issues need to be addressed:

1. *Compared with other conventional methods for statistical based clustering of gene expression profile, or biological driven protein-protein interaction network-based clustering, what is the performance gain of the bi-clustering strategy? What is the sensitivity and specificity of the method?”*

The question why we chose a biclustering approach for this study is indeed a fundamental one and we have revised the manuscript to motivate this choice more clearly. Most importantly, biclustering simultaneously clusters drugs and genes, allowing us to gain insights into both, drug action and the structure of transcriptional responses to drug perturbations within the same methodological framework. This dual clustering strategy is novel for these data sets (Lamb et al., Science, 2006). Comparative assessments have previously shown that the ISA algorithm, which we used in this study, generates highly accurate results especially for noisy biological data (Bergmann et al., Phys Rev E Stat Nonlin Soft Matter Phys, 2003; Prelic et al., Bioinform, 2006).

Action taken: Better highlighted the rationale for choosing this unsupervised biclustering algorithm and included additional citations (page 5, lines 121-123).

As an unsupervised approach, the ISA biclustering method can be applied even if a gold standard training and evaluation set is not available. This is advantageous, as, to our knowledge, gold standard data of sufficient quality on drug action at the transcriptional level do not exist. In some cases however, it appears possible to link modules to known mechanisms of drug action, allowing us to attempt to benchmark their accuracy. E.g. for the modules shown in Fig. 5 we can use the annotated mechanism of drug action (ATC code) assuming that these are “the truth” for the sake of evaluation. For modules PC3-9 (ATC code: A10B(B|G)), HL60-17 (ATC code: R03(A|B|C)(A|B|C)) and MCF7-9 (ATC code: G03) shown in Fig. 5 we estimated the sensitivity to be 0.56, 0.7 and 0.56, and the false discovery rate to be 0.58, 0.68 and 0.55, respectively. However, even from our incomplete validation experiments (we targeted only 10 out of 41 potential false positives), we can conclude that these evaluations are highly unreliable: Out of ten novel predictions (i.e. false positives in the above evaluation), we demonstrated that four of them are correct (i.e. actual true positives), suggesting that false discovery rate estimates might be exaggerated by a factor of about two (assuming that the validation rates can be generalized). This essentially confirms that a high-quality gold standard is not available. Therefore, we are not convinced that such evaluations are reliable enough (not least because of very small numbers) to add substantial value to the manuscript.

Action taken: Conducted evaluations on modules for which we attempted validation experiments, but did not include them in the revised manuscript due to extreme uncertainties associated with these estimates.

2. *“Authors should discuss the limitations of the bi-clustering strategy. For example, bi-clustering clusters drugs and genes in disjointed set. It may miss an important aspect of drug actions, i.e drugs may modulate multiple pathways. One example is cox-2 inhibitors that impact both cox-2 dependent and independent pathways.”*

We believe that this is a misunderstanding: our approach does allow each drug and each gene to be part of several modules, something we stated in the third paragraph of the first Results section (first lines of p. 6). We agree with the reviewer that this is a crucial feature to model drug actions affecting multiple pathways or polypharmacology where drugs promiscuously bind many targets and thus are likely to perturb multiple transcriptional modules.

Action taken: Rephrased corresponding paragraph to improve clarity (page 6, lines 134-138).

3. *“Majority of experimentally valid bindings are quite weak. It implies either these chemicals are promiscuity binders or the predicted off-target has little impact on the phenotype than its primary target. If it is the first case, there are many undetected false negatives. For the second case, it makes more sense that the drug is clustered with its high-affinity primary binders. One of such cases is zaprinast whose primary target is PDE. If the clustering does not detect the primary target, I think it should be considered as a false positive. The use of this drug for repurposing may be unjustified, as it may induce dominant phenotype response through its much more strongly bound primary target.”*

A similar concern was raised by reviewer #2 that the experimentally determined affinities are too weak to conclude that these drugs would be potent against the respective targets without modifications. We have adjusted the discussion of these results to more carefully state that these may nevertheless be valid as leads against the new targets which may need further optimization. Due to reduced risks and costs associated with developing such leads further, rather than starting from scratch, approaches for selective optimization of side activities (SOSA) have been developed that aim to optimize selective binding to the desired off-target over the original main activity (see e.g. Wermuth J Med Chem 2004 and references therein).

Action taken: Revised manuscript where the value of these predictions for repositioning is discussed and added references (pages 12, 13, 14 and lines 297-298, 311-312, 328-332, included additional references).

4. *“Genetic circuit is dynamic and context-dependent. The definition of conserved gene/drug modules only makes sense when the primary drug modulation pathway(s) have not been significantly rewired. Thus, it may important to take the background expression profiles into account when compare the modules derived from their differential changes.”*

It is a valid remark that drug responses occur in a dynamic and context-dependent manner. As to the first part, the dynamics of expression response, we are unfortunately limited by the fact that the main data set used, the CMap expression resource, does not contain time-resolved data (just a single time point 6h after treatment for the vast majority of chemicals). Hence our modules inherit this fundamental limitation. We fully agree that analyzing the expression dynamics of drug responses with future data sets may be extremely promising for a deeper understanding of drug action.

As to the second part of the reviewer’s remark, we are well aware of the context-dependence of these modules and therefore identified modules independently in each cell line and rat liver in order to be able to subsequently compare them. This is in contrast to most of the previous studies analyzing this data set, as these have generally averaged the signal over different cell-lines and thus essentially discarded any contextual information.

Concerning comparisons to the background cellular state: Our approach takes into account background expression patterns (a) to correct for batch effects (see Methods and Iskar et al., PLoS Comput Biol, 2010) and (b) by identifying modules among genes that change in expression when treated with drugs in comparison to the unspecific background (mean) response in each batch. We moreover filter drug-induced transcriptional modules to exclude modules which show strong responses even in the absence of drug treatments (see new Suppl. Table 2).

Action taken: Clarified in the Methods section (pages 15, 16 and lines 370-371, 401).

“More cautions should be taken when comparing in vitro modules and in vivo modules. In addition to the issue of network rewiring, pharmacokinetics may play critical roles in drug action in vivo. Moreover, dramatically different from in vitro drug binding, the drugs are rarely in an equilibrium state in vivo. Thus the determinant factor of drug-target interaction in vivo is drug resident time, instead of binding affinity measured by IC50 or Ki. I am not convinced by authors argument that drug induced gene modules are extensively conserved across cell lines, species, and in vitro and in vivo conditions.”

The reviewer is right to point out that there are many caveats to any comparison between *in vitro* (human cell lines) and *in vivo* (rat liver) data. The corresponding section in the manuscript already mentioned several of these, but we have revised it to specifically include the reviewer’s suggestions. However, even in the absence of complete data about pharmacokinetics and how it differs between model systems and despite all other biological and technical dissimilarities between data sets, we believe that it is worthwhile to identify similarities in transcriptional modules, which reflect drug-response pathways conserved at the cellular level. Importantly, similarities between the different data sets described in the manuscript were clearly detectable and significant as shown with rigorous statistics. Contrary to the reviewer, we are convinced that, because of all the differences between data sets, our conservation estimate is very likely to be overconservative (rather than spuriously inflated). How much we underestimate the true extent is difficult to say with the current data sets (our methodology applied to future data sets may help to resolve this). We have therefore modified the discussion of these results to better reflect the uncertainties in this estimate.

Action taken: Revised manuscript to add caveat of differing pharmacokinetics and to better explain why we are likely to underestimate the true extent of this phenomenon (pages 7, 14, lines 167-168, lines 351-353).

“Other issues:

1. During the bi-clustering stage, there are a number of pre-defined threshold. What is the rationale in choosing these thresholds, and how sensitive of the bi-clusters to these thresholds?”

These are very valid questions and a similar concern was also raised by reviewer #2. We thoroughly addressed them in the revised manuscript. Please see our answers to reviewer #2 above for details.

Action taken: Conducted parameter robustness analysis (new Suppl. Fig 2 and Suppl. Tables 1 and 2) and included necessary pointers in the manuscript (page 5, line 127).

“2. It is not clear how to process the time-course gene expression data of rat liver model. Although the authors provide a reference, a brief summary may be useful for readers.”

We apologize for the lack of clarity with respect to the processing of multiple time points and doses in the rat liver data. These were simply treated as independent experiments in the biclustering procedure. Information on which time points are contained in modules that were detected in rat liver and part of CODIMs is included in Suppl Table 5 (information on rat liver modules starts at p. 52 in the Supplement) and full details are provided in Supplementary Data Set 1.

Action taken: Revised Methods section to clarify how time-course data was processed (page 17, lines 407-409).